# Artificial Intelligence Approach in Machine Learning-Based Modeling and Networking of the Coronavirus Pathogenesis Pathway [note 1]

**DOI:** 10.3390/cimb47060466

**Published:** 2025-06-17

**Authors:** Shihori Tanabe, Sabina Quader, Ryuichi Ono, Hiroyoshi Y. Tanaka, Akihisa Yamamoto, Motohiro Kojima, Edward J. Perkins, Horacio Cabral

**Affiliations:** 1Division of Risk Assessment, Center for Biological Safety and Research, National Institute of Health Sciences, 3-25-26, Tonomachi, Kawasaki-ku, Kawasaki 210-9501, Japan; 2Innovation Centre of NanoMedicine (iCONM), Kawasaki Institute of Industrial Promotion, Kawasaki 210-0821, Japan; 3Division of Cellular and Molecular Toxicology, Center for Biological Safety and Research, National Institute of Health Sciences, Kawasaki 210-9501, Japan; 4Department of Pharmaceutical Biomedicine, Graduate School of Medicine, Dentistry and Pharmaceutical Sciences, Okayama University, Okayama 700-8530, Japan; 5Department of Mechanical Systems Engineering, Graduate School of Systems Design Tokyo Metropolitan University, Hachioji 192-0397, Japan; 6Department of Surgical Pathology, Kyoto Prefecture University of Medicine, Kyoto 602-8566, Japan; 7US Army Engineer Research and Development Center, Vicksburg, MS 39180, USA; 8Department of Bioengineering, Graduate School of Engineering, The University of Tokyo, Tokyo 113-0033, Japan

**Keywords:** artificial intelligence, coronavirus, coronaviral infection, machine learning, pathway analysis, prediction model, molecular network, molecular pathway image, network analysis

## Abstract

The coronavirus pathogenesis pathway, which consists of severe acute respiratory syndrome (SARS) coronavirus infection and signaling pathways, including the interferon pathway, the transforming growth factor beta pathway, the mitogen-activated protein kinase pathway, the apoptosis pathway, and the inflammation pathway, is activated upon coronaviral infection. An artificial intelligence approach based on machine learning was utilized to develop models with images of the coronavirus pathogenesis pathway to predict the activation states. Data on coronaviral infection held in a database were analyzed with Ingenuity Pathway Analysis (IPA), a network pathway analysis tool. Data related to SARS coronavirus 2 (SARS-CoV-2) were extracted from more than 100,000 analyses and datasets in the IPA database. A total of 27 analyses, including nine analyses of SARS-CoV-2-infected human-induced pluripotent stem cells (iPSCs) and iPSC-derived cardiomyocytes and fibroblasts, and a total of 22 analyses of SARS-CoV-2-infected lung adenocarcinoma (LUAD), were identified as being related to “human” and “SARS coronavirus 2” in the database. The coronavirus pathogenesis pathway was activated in SARS-CoV-2-infected iPSC-derived cells and LUAD cells. A prediction model was developed in Python 3.11 using images of the coronavirus pathogenesis pathway under different conditions. The prediction model of activation states of the coronavirus pathogenesis pathway may aid in treatment identification.

## 1. Introduction

The COVID-19 pandemic, caused by severe acute respiratory syndrome coronavirus 2 (SARS-CoV-2), has raised challenges in discovering new therapeutic agents and understanding the molecular mechanisms of emerging diseases. The coronavirus pathogenesis pathway illustrates how SARS coronavirus infection triggers cellular reactions leading to SARS coronavirus replication, adaptive immunity, innate immunity, apoptosis, lung fibrosis, acute respiratory distress syndrome (ARDS), and endothelial cell dysfunction [1,2,3]. In the coronavirus pathogenesis pathway, several signaling pathways, such as the mitogen-activated protein kinase (MAPK) pathway, the apoptosis pathway, the unfolded protein response, and the interferon (IFN) type I pathway, are activated [3]. The coronavirus pathogenesis pathway defined with Ingenuity Pathway Analysis (IPA) includes various signaling pathways involving IFN type I, transforming growth factor (TGF) beta 1 (TGFβ1), the MAPK pathway, and nodes such as MAPK components (c-jun N-terminal kinase (JNK), extracellular signal-regulated kinase 1/2 (ERK1/2), and p38MAPK) and interleukin 1B (IL1B), angiotensin II receptor type I (AGTR1), and angiotensin-converting enzyme 2 (ACE2) [4,5,6,7,8]. Previous studies demonstrated that SARS-CoV-2 infection causes pulmonary disease and cardiovascular diseases [9,10]. Cardiac side effects of the mRNA-based vaccines for SARS-CoV-2 infection or coronavirus disease 2019 (COVID-19) have been reported and are of great concern [11,12]. Fatal adverse effects related to COVID-19 vaccines were investigated, and it was found that autopsy was very useful in defining the main characteristics of the vaccine-induced immune thrombotic thrombocytopenia after ChAdOx1 nCoV-19 vaccination [13]. Causality assessment of adverse events following immunization and COVID-19 vaccination is necessary to retrace the WHO guidelines adapted for COVID-19 [14]. A study has demonstrated that mRNA-based vaccines induce specific dysfunctions in isolated adult rat cardiomyocytes [15]. While post-COVID-19 syndrome, defined as the persistence or new onset of symptoms three months after the infection that leads to a significant daily life impairment [16], has correlated to the daily life impairment caused by somatic symptom disorder [17], the mechanism of the post-COVID-19 syndrome is unknown. A study demonstrates that post-COVID-19 lung fibrosis shares immunological characteristics with idiopathic pulmonary fibrosis and suggests that SARS-CoV-2 infection activates biological pathways common with idiopathic pulmonary fibrosis [18]. It is crucial to understand the mechanisms of the diseases and predict the activation states of disease pathways for the safer development of therapeutics or vaccines.

In this study, we developed a model to predict the activation states of the coronavirus pathogenesis pathway. We aimed to develop a model to predict changes in activation status based on gene expression in the coronavirus pathogenesis pathway using an artificial intelligence (AI) approach and machine learning. In our previous study, we found that the coronavirus pathogenesis pathway was activated in diffuse-type gastric cancer [19]. Diffuse-type gastric cancer is characterized by epithelial–mesenchymal transition (EMT), a cellular phenotypic transition associated with cancer metastasis and recurrence, for which the involvement of cell cycle regulation has been identified [20]. In this study, we also investigated the relationship between diffuse-type gastric cancer networks and SARS-CoV-2 analyses.

We previously conducted AI modeling on EMT and created a highly accurate prediction model of EMT regulation pathways using a commercially available, fully automated machine learning AI platform [21]. Considering the fact that this commercial AI platform is limited to licensed users, the current study aimed to create a prediction model of coronavirus pathogenesis pathway activation using the publicly available programming language Python, thereby facilitating data democratization.

## 2. Materials and Methods

### 2.1. Coronavirus Pathogenesis Pathway and the Activation Z-Score

The coronavirus pathogenesis pathway was analyzed using the Ingenuity Pathway Analysis (IPA) network pathway tool [19,22] (https://digitalinsights.qiagen.com/products-overview/discovery-insights-portfolio/analysis-and-visualization/qiagen-ipa/) (accessed on 16 May 2025). The activation z-score of the coronavirus pathogenesis pathway was calculated in the Ingenuity Knowledge Base [22]. Briefly, the activation z-score indicates the relation in the gene expression pattern of the molecules in the dataset and the pattern that is expected based on the available literature on the coronavirus pathogenesis pathway (https://qiagen.my.salesforce-sites.com/KnowledgeBase/KnowledgeNavigatorPage?id=kA41i000000L5nQCAS&categoryName=IPA) (accessed on 16 May 2025). The activation z-score is calculated in the IPA and defined for upstream regulator analysis as follows [22]:(1)$$z\left(r\right)=\frac{\sum_{\nu \epsilon \tilde{O}}w_{R}\left(r,\nu \right)s_{R}\left(r,\;\nu \right)s_{D}\left(\nu \right)}{{\left(\sum_{\nu \in \tilde{O}}{\left[w_{R}\left(r,\nu \right)\right]}^{2}\right)}^{1/2}}$$
where(2)$$\tilde{O}\left(r\right)\;∶=\left\{\nu \in R\left(r\right)|s_{R}\left(r,\nu \right)\neq 0\wedge \;\nu \in D\right\}$$(3)$$R\left(r\right)\;∶=\left\{\nu \in V_{rg}|{\exists e\in E}_{t}\;:\;r=\sigma \left(e\right)\;\wedge \;\nu =\tau \left(e\right)\right\}$$(4)$$s_{R}\left(r,\nu \right)\;∶=s\left(e\right)\mathrm{where}\;r=\sigma \left(e\right)\;\wedge \;\upsilon =\tau \left(e\right)\mathrm{and}\;e\in \;E_{t}$$(5)$$w_{R}\left(r,\nu \right)\;∶=w\left(e\right)\mathrm{where}\;r=\sigma \left(e\right)\;\wedge \;\upsilon =\tau \left(e\right)\mathrm{and}\;e\in \;E_{t}$$

All differentially expressed genes in a given dataset that are also present as nodes in the master network form a subset $D\;\subset \;V_{g}$. Each gene in the dataset, d∈ D, can be either up- or downregulated, with the gene represented by the sign $s_{D}\left(d\right)\in \;\left\{-1,\;1\right\}$ [22]. The causal analysis algorithms are based on a “master” network, which is derived from the Ingenuity Knowledge Base, and given by a directed multigraph $G=\left(V,\;E\right)$, with nodes υ∈ *V* representing mammalian genes, chemicals, protein families, complexes, microRNA species, and biological processes, and edges e∈E reflecting observed cause–effect relationships [22]. Edges are also associated with weights $w\left(e\right)\in \left\{0,\;1\right\}$, reflecting the confidence in the assigned direction of the effect [22]. Further details are provided in the work of Kämer et al. [22].

A network is a graphical representation of the molecular relationships between molecules. Molecules are represented as nodes, and the biological relationship between two nodes is represented as an edge (line). All edges are supported by at least one reference from the literature, from a textbook, or from canonical information stored in the QIAGEN Knowledge Base. The intensity of the node color indicates the degree of up-(red) or down-(green) regulation. Nodes are displayed using various shapes that represent the functional class of the gene product (https://qiagen.my.salesforce-sites.com/KnowledgeBase/KnowledgeNavigatorPage?id=kA41i000000L5pXCAS&categoryName=IPA) (accessed on 16 May 2025).

### 2.2. Network Analysis

As of 2021, 106 analyses and 106 datasets from more than 100,000 pieces of data were found to be related to SARS-CoV in the IPA database. We filtered the 106 analyses associated with “SARS coronavirus 2” with the term “human,” with a total of 49 analyses being identified. The 49 analyses consisted of 22 analyses of lung adenocarcinoma (LUAD) and 27 analyses of non-LUAD data. The gene expression data of the analyses are publicly available in Gene Expression Omnibus (GEO) as GEO Series (GSE) (https://www.ncbi.nlm.nih.gov/geo/) (accessed on 16 May 2025). The data from the 22 analyses of LUAD included 13 analyses of LUAD infected with SARS-CoV-2 of the amlodipine series (GSE154613) [23], one analysis of LUAD bronchial epithelial Calu-3 cells infected with SARS-CoV Urbani strain (GSE17400) [24], and eight analyses of LUAD cells infected with SARS-CoV-2 (GSE147507) (ACE2-transfected A549 cells) [25,26]. We analyzed eight analyses of GSE147507 [25,26] and the data on LUAD bronchial epithelial cells infected with SARS-CoV (GSE17400) [24]. The 27 analyses of non-LUAD data included nine analyses of induced pluripotent stem cell (iPSC)-derived cardiac cells (GSE156754) [9,27].

### 2.3. Analysis Match

SARS-CoV-2 analyses data and diffuse-type gastric cancer data were compared using the Analysis Match function in the IPA database (https://qiagen.my.salesforce-sites.com/KnowledgeBase/KnowledgeNavigatorPage?id=kA41i000000L5vUCAS&categoryName=IPA) (accessed on 16 May 2025). Our previous finding showed a certain degree of correlation between diffuse-type gastric cancer data and the virus infection pathway [28]. The Analysis Match function identified the entities related to diffuse-type gastric cancer and SARS-CoV-2 datasets. The entities of the Analysis Math included upstream regulators (URs), master regulators in causal networks (CNs), and diseases and functions in downstream effects (DEs).

### 2.4. Activity Plot Analysis

The activity plot of the activation z-score of the coronavirus pathogenesis pathway in the IPA database identified more than 10,000 analyses as of 2021, of which 100 (50 activated and 50 inactivated) pathway maps of the coronavirus pathogenesis pathway were obtained and used as training data for machine learning. The details of training data are described in Section 2.5, Python Coding.

### 2.5. Python Coding

We created a model to predict the activation state using Python (GitHub: https://github.com/Shihori/AI/blob/68bbeaf7944ff8d1c3bbcb40fbf011c853cf4383/CNN_corona2_GitHub.ipynb) (accessed on 16 May 2025). The Python code was created with reference to the textbook “Machine Learning in Python: Machine Learning of Life Science Data” [29]. The Python code for convolutional neural network modeling, detailed in section six in the textbook, was used to create the prediction model of the activation states of the coronavirus pathogenesis pathway [29]. The 100 images of the coronavirus pathogenesis pathway (50 activated and 50 inactivated) obtained in the network pathway analysis with IPA were uploaded, among which 70, 20, and 10 images were used as training, validation, and test datasets in Google Colaboratory (binary classification). The model validation methodology is based on a split between training and test datasets. The Gradient-weighted Class Activation Mapping (Grad-CAM) technique was used to interpret the decision of the AI in distinguishing image activation by coloring the locus of the AI’s attention [29]. Subsequently, the model VGG16 for transfer learning was programmed in Python 3.11 (GitHub: https://github.com/Shihori/AI/blob/68bbeaf7944ff8d1c3bbcb40fbf011c853cf4383/CNN_corona2_GitHub.ipynb) (accessed on 16 May 2025) [29].

### 2.6. Statistical Analysis

The RNA sequencing data on diffuse-type gastric cancer were analyzed with IPA, as previously described [20]. The activation z-score in each network or pathway was calculated with IPA to show the level of activation [22].

## 3. Results

### 3.1. Molecular Network Analysis of SARS-CoV-2

The LUAD data were compared and analyzed with IPA because, as of 2011, 22 LUAD analyses related to the terms “human” and “SARS coronavirus 2” have been identified when searching with this form of analysis, and lung fibrosis was included in the coronavirus pathogenesis pathway. The data from the 22 analyses included 13 analyses of GSE154613 (amlodipine series) [23], one analysis of GSE17400 (Calu-3 cells infected with SARS-CoV Urbani strain) [24], and eight analyses of GSE147507 (ACE2-transfected A549 cells) [25]. The activation z-score of the coronavirus pathogenesis pathway in eight analyses of SARS-CoV-2-infected A549 cells in 0.2 or 2 multiplicity of infection (MOI) (GSE147507) is shown in Table 1. The heatmap of activation states of canonical pathways of the eight analyses of SARS-CoV-2-infected A549 cells (GSE147507) as of 2024 is shown in Figure 1.

### 3.2. Coronavirus Pathogenesis Pathway in LUAD Samples Infected with SARS-CoV

The coronavirus pathogenesis pathway was overlaid with data from the LUAD samples infected with SARS-CoV (Figure 2). The coronavirus pathogenesis pathway was overlaid with gene expression data of LUAD bronchial epithelial cell (Calu-3 cell line) samples infected with SARS-CoV Urbani strain in 0.1 MOI compared to mock-infected Calu-3 cells (GSE17400) (as of 2024).

### 3.3. SARS-CoV-2 Analysis Matched with Diffuse-Type Gastric Cancer

Another dataset related to SARS-CoV-2 included data from iPSCs infected with SARS-CoV-2. The findings of previous studies revealed a certain degree of correlation between a network of diffuse-type gastric cancer and RNA viral infection [19,28]. To further investigate the regulation mechanism of SARS-CoV-2-related networks and diffuse-type gastric cancer, a series of SARS-CoV-2-related analyses were compared to the analyzed gene expression data of diffuse-type gastric cancer. The five analyses related to SARS-CoV-2 (iPSC-derived cardiomyocyte infected with SARS-CoV-2 0.001 MOI vs. mock, iPSC-derived cardiomyocyte infected with SARS-CoV-2 0.01 MOI vs. mock, iPSC-derived cardiomyocyte infected with SARS-CoV-2 0.1 MOI vs. mock, iPSC-derived cardiac fibroblast infected with SARS-CoV-2 0.006 MOI vs. mock, and iPSC infected with SARS-CoV-2 0.006 MOI vs. mock) (GSE156754 https://www.ncbi.nlm.nih.gov/geo/query/acc.cgi?acc=GSE156754) (accessed on 16 May 2025) [9,27] were compared with the analysis result of gene expression data of diffuse-type gastric cancer using the Analysis Match function in IPA (as of 2022). The compared SARS-CoV-2 data comprised the RNA sequencing data of iPSCs or iPSC-derived cardiac cells infected with SARS-CoV-2 (GSE156754) in the IPA database. The upstream regulators (URs) identified in the analyses included TP53, let-7, CDKN2A, calcitriol, NUPR1, SMARCB1, MEF2D, decitabine, SPARC, and RB1 (Table 2). A similar activation z-score was observed in let-7 when the data of diffuse-type gastric cancer were compared to SARS-CoV-2 analyses in the IPA. The entity types identified included upstream regulators (URs), master regulators in causal networks (CNs), and diseases and functions in downstream effects (DEs) (Table 2).

### 3.4. Coronavirus Pathogenesis Pathway in Stem Cells

The coronavirus pathogenesis pathway was overlaid with the analysis of SARS-CoV-2-infected skin-derived iPSCs (GSE156754) [9,27] in the IPA database. The RNA sequencing data of iPSCs infected with 0.006 multiplicity of infection (MOI) of SARS-CoV-2 were compared to the data of iPSCs infected with mock (Figure 3). In the analysis, the expression of TP53, STAT3, JAK1, STAT2, EP300, ATP6AP1, DPP9, ZC3HAV1, and NUP98 was upregulated in iPSCs infected with SARS-CoV-2 compared to the mock infection (Figure 3).

### 3.5. Drugs That Interact with the Coronavirus Pathogenesis Pathway

Drugs that interact with the nodes in the coronavirus pathogenesis pathway were analyzed in the IPA network pathway analysis tool. The nodes in the coronavirus pathogenesis pathway, which are the target of drugs, are summarized in Table 3. The drugs interacting with the nodes in the coronavirus pathogenesis pathway included telmisartan, acetaminophen, and arsenic trioxide (Table 3). The drugs that have more than three target molecules are overlaid with the coronavirus pathogenesis pathway and colored by the expected activation state in Figure 4.

### 3.6. Prediction Modeling of the Activation States of Coronavirus Pathogenesis Pathway (Python Modeling)

In our previous study, we established AI models related to epithelial–mesenchymal transition (EMT) using the DataRobot platform [21]. We generated a model using Python coding in this study. The Python code for convolutional neural network modeling was used to create the prediction model of the activation states of the coronavirus pathogenesis pathway. The data used to formulate the prediction model included 100 images (50 activated and 50 inactivated images) of the coronavirus pathogenesis pathway (binary classification) (Figure 5a). Among the 100 images, 70, 20, and 10 were used as training, validation, and test datasets. The accuracy of the model tested with 10 datasets was 0.3 before transfer learning, which increased to 0.65 after transfer learning. The Grad-CAM technique was used to interpret the AI’s decision, distinguishing the activation of the images by coloring the locus of the AI’s attention. Subsequently, the model VGG16 for transfer learning was programmed in Python. The transfer learning model in Python was found to have an accuracy of 65.0% (Figure 5b). The code is available through GitHub: https://github.com/Shihori/AI/blob/68bbeaf7944ff8d1c3bbcb40fbf011c853cf4383/CNN_corona2_GitHub.ipynb (accessed on 16 May 2025).

The convolutional neural network model was compared to a graph neural network to demonstrate the significance of machine learning with images (https://github.com/Shihori/AI/blob/018d5cfa90265cf00c6b5cf2a66cd8a2a8009b50/GNN_corona2_GitHub_py.ipynb) (accessed on 16 May 2025). The graph neural network code was created by generative AI, Google Gemini 2.5 Flash. The accuracy of the graph neural network model was 0.4, which highlighted the significance of the transfer learning model in Python.

## 4. Discussion

In the coronavirus pathogenesis pathway, the TGFβ1-SMAD3 pathway leading to lung fibrosis is activated during coronavirus infection. The NFκB, FOS, and JUN pathways are activated, leading to hypercytokinemia and T-cell dysfunction. The gene expression of IL6 and CXCL8 is elevated during coronavirus infection. Interferon (IFN) type I signaling, leading to adaptive immunity, is also involved in the coronavirus pathogenesis pathway.

A model to predict the activation states of the coronavirus pathogenesis pathway using images was generated in the current study using Python. The activation state of samples analyzed with IPA differs under several different conditions. We previously created the model using a commercially available, fully automated machine learning AI platform. Although Python coding successfully predicted some of the data images, its accuracy is far from that of the commercially available, fully automated machine learning platform. Additional refinement of the model in Python is necessary in order to achieve higher accuracy. One of the factors responsible for the low accuracy of the current model may be the fact that the binary classification approach was used to train the model. A regression model involving the use of continuous values of activation z-score may improve the accuracy of the prediction model. The results of the Grad-CAM model may even aid in the identification of the specified locus of attention if the accuracy of the model is improved. Using the Grad-CAM model may be helpful for identifying the specific therapeutic targets of drugs. The activation state of a pathway is determined based on changes in signal intensity in the current study. The purpose of using “images” colored by changes in signal intensity and predicted activation state is that humans can easily interpret the pathway images with colors. The ultimate goal is to predict the activation state of a pathway based on changes in the expression of nodes in the path. The graph neural network model had lower accuracy compared to the convolutional neural network model, highlighting the significance of machine learning with images. Data augmentation by modifying the location of nodes in pathway images may also be a future investigation to enhance the accuracy of the model.

Although IPA is a widely used pathway analysis tool, the data sources and algorithms are proprietary, which may introduce potential biases in the prediction models created. Using other pathway analysis tools, such as Reactome (https://reactome.org) (accessed on 16 May 2025) or Kyoto Encyclopedia of Genes and Genomes (KEGG) (https://www.kegg.jp) (accessed on 16 May 2025), would be a future investigation [30,31]. The mechanistic validation of the predicted results is essential for the application of the AI-based machine learning model in clinical use. It would be crucial to have experimental evidence to confirm the biological validity of the novel targets of the coronavirus pathogenesis pathway.

Previous studies have suggested that the molecular network of gastric cancer and the RNA virus infection network interact [19,28]. Upstream regulators identified in the comparison of the diffuse-type gastric cancer and SARS-CoV-2 data included TP53, let-7, SMARCB1, MEF2D, decitabine, and SPARC, among which let-7 was activated in both diffuse-type gastric cancer and SARS-CoV-2, whereas TP53, SMARCB1, MEF2D, decitabine, and SPARC were activated in diffuse-type gastric cancer and inactivated in SARS-CoV-2. Decitabine is a nucleoside metabolic inhibitor approved in the United States and indicated for the treatment of adult patients with myelodysplastic syndromes that targets DNA methyltransferase (DNMT) [32,33]. The difference between the activation states of diffuse-type gastric cancer and SARS-CoV-2 may provide clues for the treatment of coronaviral diseases. The activation state of let-7 was activated in both diffuse-type gastric cancer and SARS-CoV-2. The authors of a previous study found that a small molecule, C1632, inhibits SARS-CoV-2 replication by blocking the interaction between LIN28 and pri/pre-let-7 to promote the maturation of let-7 [34]. It has been established that let-7 directly inhibits IL-6 expression [35]. IL-6 levels are significantly elevated in COVID-19 patients [36]. These insights identified in the network analyses of SARS-CoV-2 and diffuse-type gastric cancer samples may aid in treatment identification.

## 5. Conclusions

In conclusion, we developed a model for predicting the activation state of the coronavirus pathogenesis pathway using a Python approach [33]. The accuracy of the model with transfer learning was 65.0%. The limitation of our study is that the accuracy of the prediction model needs to be improved. The newly developed model for predicting the activation state of the coronavirus pathogenesis pathway may aid in predicting the responsiveness of drugs to treat diseases caused by new coronaviruses. In the future, it will be necessary to study the relationship between the molecular network activation state and the pathological mechanism to find therapeutic approaches for the treatment of various diseases.

## Figures and Tables

**Figure 1 cimb-47-00466-f001:**
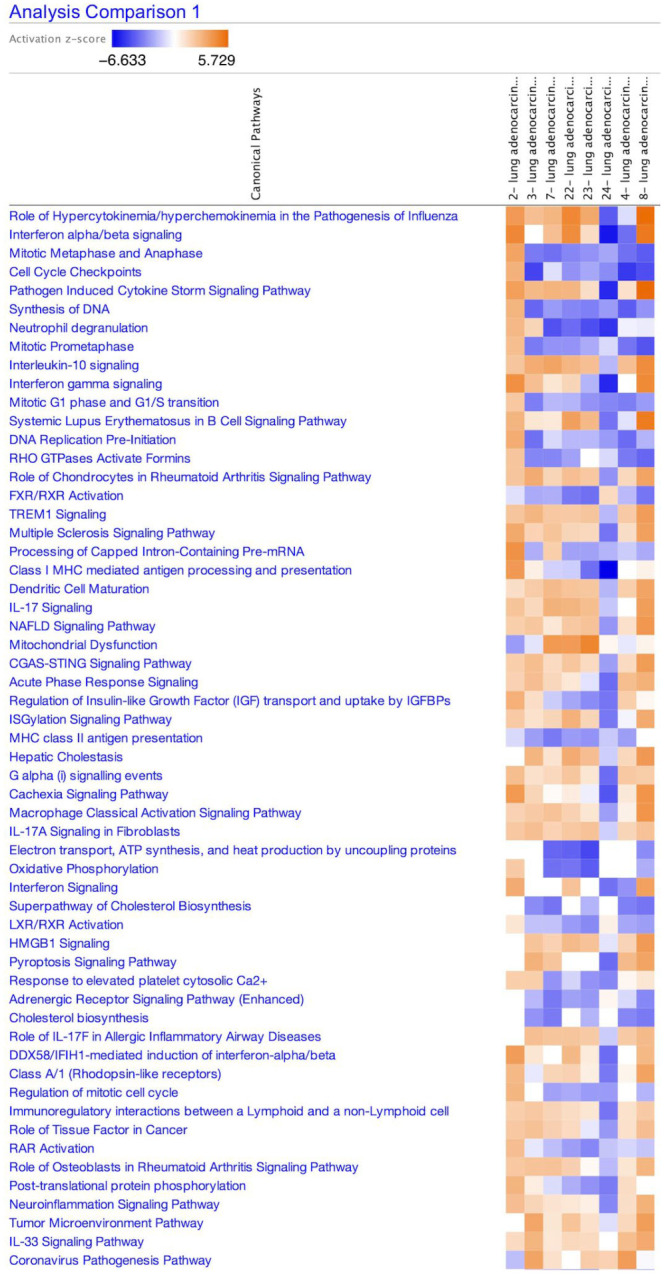
SARS coronavirus 2 data analysis. The identified canonical pathways of SARS-CoV-2-infected A549 cells (LUAD) (ACE2 transfection) (GSE147507) (eight analyses) in comparison analysis are shown. The coronavirus pathogenesis pathway is located at the bottom of the heatmap. Orange or blue coloring indicates activation or inactivation in the analysis. The analysis names are indicated in Table 1.

**Figure 2 cimb-47-00466-f002:**
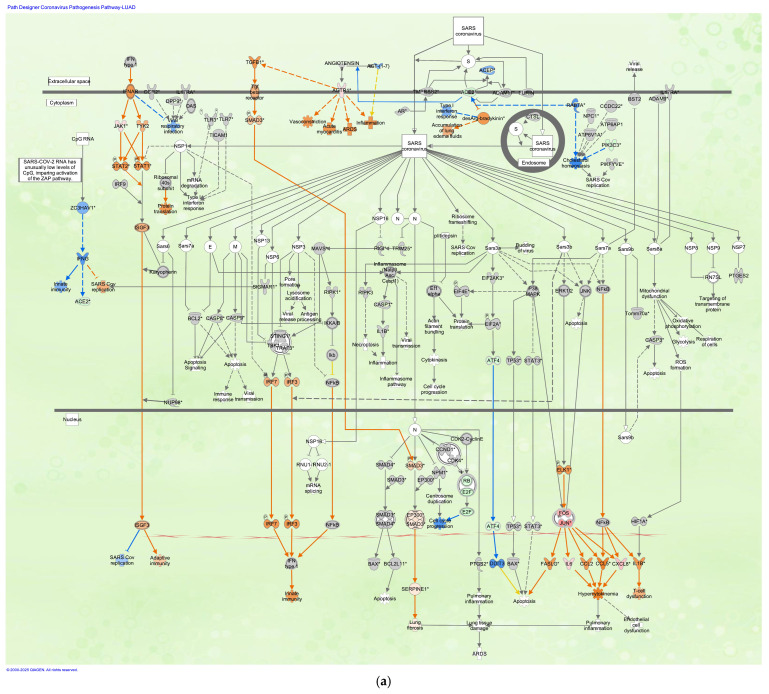
Coronavirus pathogenesis pathway: (**a**) The coronavirus pathogenesis pathway was overlaid with LUAD (bronchial epithelium) infected with SARS-CoV (As of April 2024) (SARS-CoV data, GSE17400). (**b**) The prediction legend of the pathway is shown. Red or green coloring indicates upregulated or downregulated gene expression, respectively. Orange or blue coloring indicates predicted activation or inhibition, respectively. The intensity of the colors indicates the degree of up- or downregulation. An orange or blue line indicates activation or inhibition, respectively. (**c**) The legend for the node shapes in the pathway is shown. (**d**) The legend for the relationship lines is also shown. Gene/Protein/Chemical identifiers marked with an asterisk indicate that multiple identifiers in the dataset file map to a single gene/chemical in the Global Molecular Network.

**Figure 3 cimb-47-00466-f003:**
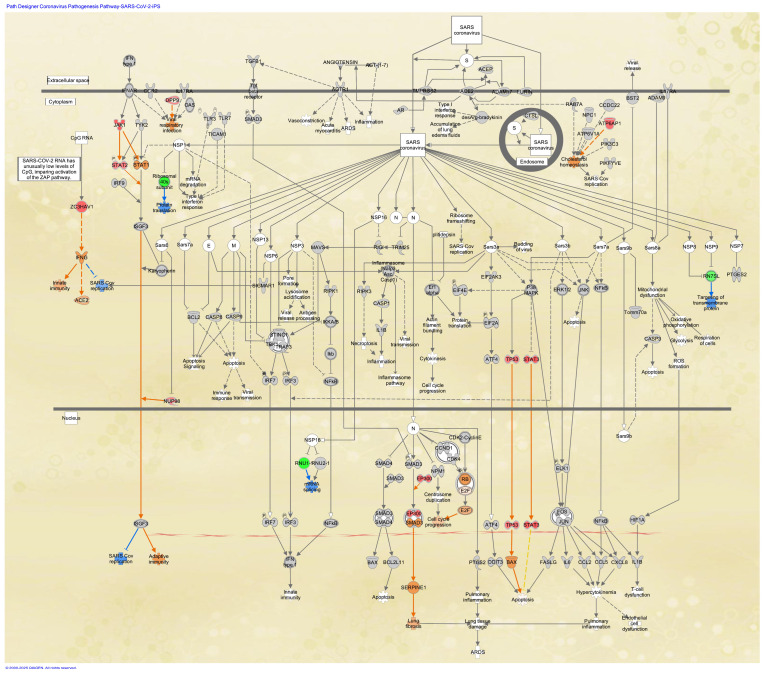
Coronavirus pathogenesis pathway overlaid with gene expression data of iPSCs. The coronavirus pathogenesis pathway was overlaid with iPSCs infected with SARS-CoV-2. The expression log ratio of iPSCs infected with SARS-CoV-2 (0.006 MOI) to mock infection was colored in red (upregulated) or green (downregulated) (as of April 2024). Gene expression data (GSE156754) of iPSCs with mock infection and iPSCs infected with SARS-CoV-2 were compared in the IPA database. Red or green coloring indicates upregulated or downregulated gene expression, respectively. Orange or blue coloring indicates predicted activation or inhibition, respectively. The intensity of the colors indicates the degree of up- or downregulation. An orange or blue line indicates activation or inhibition, respectively.

**Figure 4 cimb-47-00466-f004:**
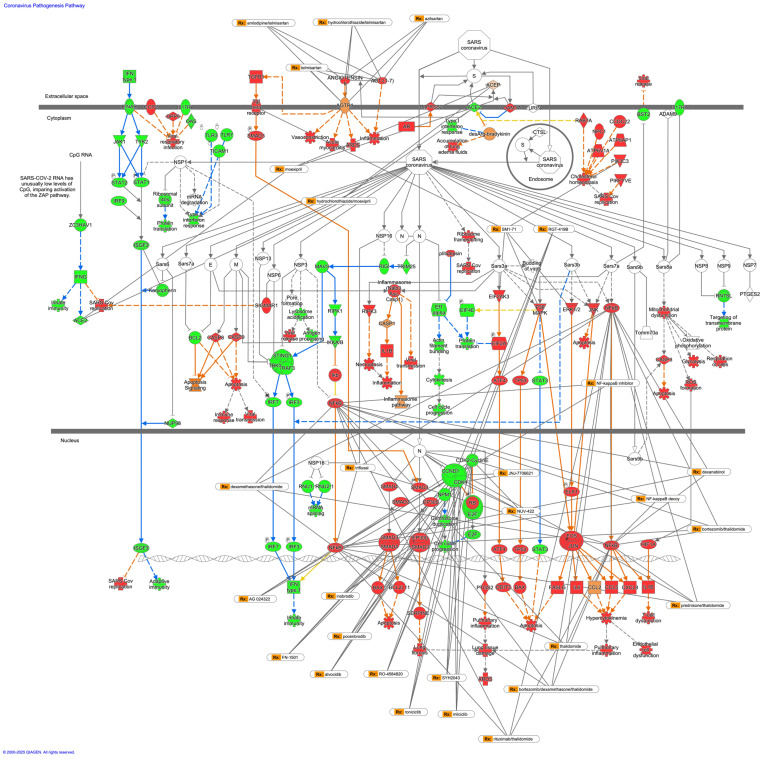
Coronavirus pathogenesis pathway overlaid with the drugs that have more than three target molecules. The coronavirus pathogenesis pathway was colored by expected activation state. Red or green coloring indicates upregulated or downregulated activation state, respectively. Orange coloring indicates predicted activation. An orange or blue line indicates activation or inhibition, respectively.

**Figure 5 cimb-47-00466-f005:**
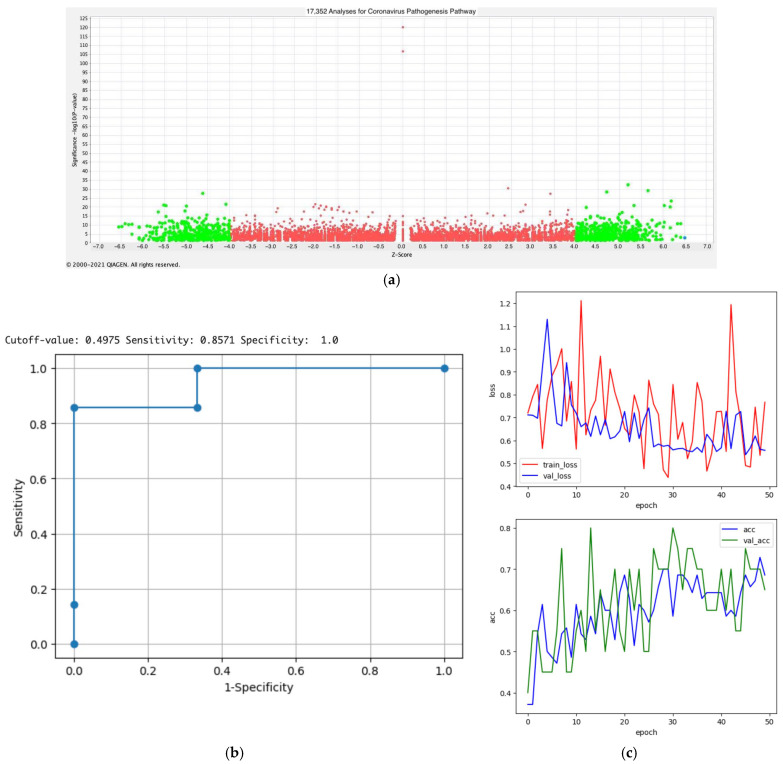
Python modeling of the activation states of the coronavirus pathogenesis pathway. (**a**) Activity plot of 17,352 analyses for the coronavirus pathogenesis pathway is shown (as of 2021). The x-axis indicates the z-scores of the analyses. The higher score is more activated, while the lower score is less activated. Analyses with an absolute z-score of more or less than 4 are indicated in green or red. The analysis with the highest z-score is indicated in a blue dot. The 50 activated and 50 inactivated coronavirus pathogenesis pathway images were used to train the model. (**b**) The ROC curve of the prediction result of the test data is shown. (**c**) The results of the prediction model for the activation states of the coronavirus pathogenesis pathway are shown. The model with transfer learning with Python was found to have an accuracy of 65.0%. In the graphs, “train,” “acc,” and “val” indicate “training,” “accuracy,” and “validation,” respectively. The corresponding code is available through GitHub: https://github.com/Shihori/AI/blob/68bbeaf7944ff8d1c3bbcb40fbf011c853cf4383/CNN_corona2_GitHub.ipynb (accessed on 16 May 2025).

**Table 1 cimb-47-00466-t001:** Activation states of the coronavirus pathogenesis pathway of SARS-CoV-2-infected A549 cells.

Analysis Name	Activation z-Score of Coronavirus Pathogenesis Pathway *	Comparison Contrast
2-lung adenocarcinoma (LUAD) alveoli 7103	−1.706	SARS-CoV-2-infected A549 cell line (MOI 0.2) vs. mock-infected A549 cell line
3-lung adenocarcinoma (LUAD) alveoli 7109	3.464	SARS-CoV-2-infected A549 cell line (MOI 2) vs. mock-infected A549 cell line
7-lung adenocarcinoma (LUAD) alveoli 7113	1.147	SARS-CoV-2-infected ACE2-transfected A549 cell line (MOI 0.2) vs. mock-infected ACE2-transfected A549 cell line
22-lung adenocarcinoma (LUAD) alveoli DMSO 7106	0	SARS-CoV-2-infected ACE2-transfected A549 cell line vs. mock-infected ACE2-transfected A549 cell line
23-lung adenocarcinoma (LUAD) alveoli ruxolitinib 7107	1.941	SARS-CoV-2-infected ACE2-transfected A549 cell line and ruxolitinib vs. mock-infected ACE2-transfected A549 cell line
24-lung adenocarcinoma (LUAD) alveoli ruxolitinib 7108	1.732	SARS-CoV-2-infected ACE2-transfected A549 cell line and ruxolitinib vs. SARS-CoV-2-infected ACE2-transfected A549 cell line
4-lung adenocarcinoma (LUAD) alveoli 7110	3.742	SARS-CoV-2-infected A549 cell line (MOI 2) vs. SARS-CoV-2 infected A549 cell line (MOI 0.2)
8-lung adenocarcinoma (LUAD) bronchial epithelium 7114	−0.2	SARS-CoV-2-infected CALU3 cell line vs. mock-infected CALU-3 cell line

* The activation z-score indicates the activation state of the pathway. A positive or negative score indicates activation or inactivation, respectively. Further details are provided in the Section 2.

**Table 2 cimb-47-00466-t002:** SARS-CoV-2 analysis matched to diffuse-type gastric cancer and the activation z-score.

Entity Type	Entity Name	Diffuse-Type Gastric Cancer	iPSC-Derived Cardiomyocyte Infected with SARS-CoV-2 0.001 MOI vs. Mock	iPSC-Derived Cardiomyocyte Infected with SARS-CoV-2 0.01 MOI vs. Mock	iPSC-Derived Cardiomyocyte Infected with SARS-CoV-2 0.1 MOI vs. Mock	iPSC-Derived Cardiac Fibroblast Infected with SARS-CoV-2 0.006 MOI vs. Mock	iPSC Infected with SARS-CoV-2 0.006 MOI vs. Mock
DE	Organismal death	6.09939477	0	0	0	0	−10.332512
DE	Morbidity or mortality	6.0991701	0	0	0	0	−10.254265
UR	TP53	5.25209454	−2.9576275	−3.8544245	0	−4.0046362	3.36923989
UR	let-7a-5p (and other miRNAs w/seed GAGGUAG)	2.95701052	2.98384345	3.24713706	2.75140771	0	−3.6818652
UR	let-7	5.88141247	3.36872653	3.07534027	2.76863583	−0.7453134	−2.628098
UR	CDKN2A	5.00037308	0.34050945	0.51898468	1.34164079	−3.2237322	2.97677657
UR	calcitriol	5.35668014	0	0	1.23787842	−1.587867	1.9593573
CN	NUPR1	6.68503217	0	0	0	−6.0621778	0.33752637
CN	l-asparaginase	7.00201178	0	0	0	−4.2	0
UR	l-asparaginase	6.92462738	0	0	2.23606798	−4.1949137	0
UR	NUPR1	6.68503217	0	0	2.49615088	−6.0621778	−0.386494
UR	SMARCB1	2.84931818	−1.8898224	−1.1338934	−2	−1.407767	2.57658201
UR	MEF2D	2.38560366	−2.5729119	−2.3785413	−1.9249444	−2.236068	0
UR	Decitabine	3.08835855	−3.4575395	−2.2066886	−1.5180635	0	−0.2058335
UR	SPARC	3.28571429	−1.3516756	−1.7509621	−1.9686483	0	0
DE	Growth failure or short stature	3.70765671	0	0	0	0	−5.311879
UR	RB1	3.36893187	0	0	−1.8347785	−1.3252763	−2.7095152
CN	Osimertinib	5.93335075	0	0	1	0	−2.4596748

**Table 3 cimb-47-00466-t003:** Nodes in the coronavirus pathogenesis pathway that interact with drugs.

Symbol	Entrez Gene Name	Drugs
ACE2	angiotensin converting enzyme 2	hydrochlorothiazide/moexipril, and moexipril
ACEP	angiotensin I converting enzyme	ceronapril, indolapril, pentopril, quinaprilat, perindoprilat, angiotensin I (1– 7), amlodipine/perindopril, benazeprilat, trandolaprilat, angiotensin-converting enzyme inhibitor, aspirin/lisinopril, amlodipine/benazepril, hydrochlorothiazide/lisinopril, benazepril, enalapril, perindopril, captopril, cilazapril, enalapril/felodipine, hydrochlorothiazide/moexipril, benazepril/hydrochlorothiazide, hydrochlorothiazide/quinapril, fosinopril/hydrochlorothiazide, captopril/hydrochlorothiazide, enalapril/hydrochlorothiazide, hydrochlorothiazide/trandolapril, ramipril, ramiprilat, moexipril, quinapril, amlodipine/indapamide/perindopril, lisinopril, enalaprilat, trandolapril, moexiprilat, idrapril, rentiapril, imidaprilat, gemopatrilat, zabiciprilat, libenzapril, fosinoprilat, zofenoprilat, trandolapril/verapamil, diltiazem/enalapril, fosinopril, and carvedilol/enalapril
ADAM17	ADAM metallopeptidase domain 17	aderbasib
ADAM9	ADAM metallopeptidase domain 9	IMGC936
AGTR1	angiotensin II receptor type 1	caffeine/dextromethorphan/losartan/midazolam/omeprazole, amlodipine/olmesartan medoxomil, olmesartan, amlodipine/hydrochlorothiazide/valsartan, amlodipine/telmisartan, aliskiren/valsartan, azilsartan, azilsartan kamedoxomil, amlodipine/hydrochlorothiazide/olmesartan medoxomil, aspirin/dipyridamole/telmisartan, clopidogrel/telmisartan, azilsartan medoxomil/chlorthalidone, sacubitril/valsartan, amlodipine/valsartan, sparsentan, nebivolol/valsartan, hydrochlorothiazide/losartan, hydrochlorothiazide/valsartan, candesartan, candesartan cilexetil, olmesartan medoxomil, irbesartan, losartan potassium, telmisartan, eprosartan, candesartan cilexetil/hydrochlorothiazide, hydrochlorothiazide/irbesartan, eprosartan/hydrochlorothiazide, hydrochlorothiazide/telmisartan, hydrochlorothiazide/olmesartan medoxomil, amlodipine/ezetimibe/losartan/rosuvastatin, and valsartan
ANGIOTENSINII	angiotensinogen	amlodipine/telmisartan, azilsartan, telmisartan, and hydrochlorothiazide/telmisartan
AR	androgen receptor	TAS3681, ARV-110, ODM 204, bicalutamide/GnRH analog, CC-94676, CB0310, EPI-7386, AC176, estradiol valerate/testosterone enanthate, estradiol cypionate/testosterone cypionate, ARV-766, TQB3720, BMS-641988, cyproterone acetate/ethinyl estradiol, enzalutamide, galeterone, ostarine, 1-testosterone, clascoterone, flutamide/goserelin, nandrolone phenpropionate, androgen receptor antagonist, apalutamide, darolutamide, AZD3514, APC-100, EPI-506, bicalutamide/leuprolide, bicalutamide/goserelin, dexamethasone/enzalutamide, rezvilutamide, LY2452473, enzalutamide/exemestane, drospirenone/ethinyl estradiol, nilutamide, TRC253, bicalutamide, SXL01, proxalutamide, hydroxyflutamide, testolone, GSK2881078, flutamide, nandrolone decanoate, testosterone cypionate, deuterated enzalutamide, AZD5312, fluoxymesterone/tamoxifen, cyproterone acetate, nandrolone, drospirenone, medroxyprogesterone acetate, oxandrolone, danazol, dihydrotestosterone, fluoxymesterone, stanozolol, spironolactone, methyltestosterone, testosterone, oxymetholone, 7alpha-hydroxytestosterone, norgestimate, testosterone propionate, and testosterone enanthate
ATP6V1A	ATPase H+ transporting V1 subunit A	bafilomycin A1 and bafilomycin b1
BCL2	BCL2 apoptosis regulator	BP1002, S65487, rosomidnar, FCN-338, BGB-11417, LP-118, ZN-d5, oblimersen, ABBV-623, ABBV-453, TQB3909, rasagiline, (-)-gossypol, obatoclax, ABT-737, BCL-2 blocker, navitoclax, gemcitabine/paclitaxel, bortezomib/paclitaxel, venetoclax, paclitaxel/trastuzumab, paclitaxel/pertuzumab/trastuzumab, lapatinib/paclitaxel, doxorubicin/paclitaxel, epirubicin/paclitaxel, paclitaxel/ramucirumab, paclitaxel/topotecan, BCL201, pelcitoclax, paclitaxel/rituximab, afatinib/paclitaxel, doxorubicin/lapatinib/paclitaxel/trastuzumab, levodopa/rasagiline, everolimus/paclitaxel, lisaftoclax, SPC2996, paclitaxel/pembrolizumab/ramucirumab, paclitaxel, chelerythrine, AZD0466, and paclitaxel/pembrolizumab, LP-108
CASP1	caspase 1	caspase 1 inhibitor
CASP3	caspase 3	caspase 3 inhibitor
CASP9	caspase 9	caspase-9 inhibitor
CCL2	C-C motif chemokine ligand 2	CNTO 888, mimosine
CCND1	cyclin D1	arsenic trioxide/tretinoin, arsenic trioxide/daunorubicin/tretinoin, arsenic trioxide/gemtuzumab ozogamicin/tretinoin, arsenic trioxide/idarubicin/tretinoin, arsenic trioxide/cytarabine/methotrexate, and arsenic trioxide
CCR2	C-C motif chemokine receptor 2	AZD2423, PF-4136309, MLN1202, BMS-813160, propagermanium, ilacirnon, and MK-0812
CDK4	cyclin dependent kinase 4	XZP-3287, CINK4, PD 0183812, BPI-16350, PF-07220060, dalpiciclib, TQB3616, NUV-422, TQB3303, narazaciclib, CS3002, TY-302, RGT-419B, RO0506220, palbociclib, PRT3645, SYH2043, QLS12004, SPH4336, cyclin dependent kinase 4 inhibitor, riviciclib, AG 024322, milciclib, RO-4584820, ribociclib, voruciclib, abemaciclib, letrozole/palbociclib, roniciclib, FLX925, fulvestrant/palbociclib, trilaciclib, lerociclib, letrozole/ribociclib, abemaciclib/fulvestrant, anastrozole/palbociclib, anastrozole/ribociclib, exemestane/palbociclib, exemestane/ribociclib, abemaciclib/aromatase inhibitor, JNJ-7706621, fulvestrant/ribociclib, abemaciclib/exemestane, abemaciclib/anastrozole, abemaciclib/letrozole, ribociclib/tamoxifen, everolimus/ribociclib, fascaplysin, abemaciclib/fulvestrant/GnRH analog, abemaciclib/aromatase inhibitor/GnRH analog, HS-10342, FCN-437, FN-1501, alvocidib, GLR2007, BPI-1178
CTSL	cathepsin L	pegulicianine, and cathepsin L inhibitor
CXCL8	C-X-C motif chemokine ligand 8	BMS-986253
EIF2AK3	eukaryotic translation initiation factor 2 alpha kinase 3	HC-5404-FU, NMS-03597812, AMG44, GSK2656157, and SM1-71
EIF4E	eukaryotic translation initiation factor 4E	ISIS 183750
EP300	E1A binding protein p300	pocenbrodib and inobrodib
FASLG	Fas ligand	APG101
FURIN	furin, paired basic amino acid cleaving enzyme	hexa-D-arginine, furin inhibitor, and nona-D-arginine amide
HIF1A	hypoxia inducible factor 1 subunit alpha	BAY 87-2243, PX 478, and EZN 2968
IFNG	interferon gamma	emapalumab
IL17RA	interleukin 17 receptor A	brodalumab
IL1B	interleukin 1 beta	FL-101, anakinra, rilonacept, AK114, canakinumab, gevokizumab, canakinumab/INS, canakinumab/metformin, canakinumab/metformin/sulfonylurea, canakinumab/colchicine, canakinumab/methotrexate, anakinra/methotrexate, and gallium nitrate
IL6	interleukin 6	anti-IL-6 monoclonal antibody, tocilizumab, vamikibart, siltuximab, clazakizumab, interleukin-6 receptor inhibitor, and ziltivekimab
JAK1	Janus kinase 1	ivarmacitinib, solcitinib, deuruxolitinib, delgocitinib, momelotinib metabolite M21, tofacitinib, ruxolitinib, momelotinib, baricitinib, INCB-16562, filgotinib, oclacitinib, SAR-20347, itacitinib, INCB052793, methotrexate/tofacitinib, upadacitinib, brepocitinib, abrocitinib, baricitinib/methotrexate, pralsetinib, JAK inhibitor I, JAK1 inhibitor, AZD4205, povorcitinib, erlotinib/ruxolitinib, jaktinib, tinengotinib, and methotrexate/ruxolitinib/vincristine
JNK		JNK inhibitor, CC 401, SR-3562, and AS601245
NFkB		NF-kappaB inhibitor and dexanabinol
P38MAPK		SB 220025, doramapimod, AZ10164773, PHA-666859, acumapimod, PD 169316, merck C, SC68376, SK & F 86002, SB 239063, SD-282, SB203580, RWJ 67657, and TAK715
PIK3C3	phosphatidylinositol 3-kinase catalytic subunit type 3	PIK-III, VPS34 inhibitor 1, SAR405, VPS34-IN1, and SM1-71
PIKFYVE	phosphoinositide kinase, FYVE-type zinc finger containing	APY0201 and apilimod
PTGS2	prostaglandin-endoperoxide synthase 2	bupivacaine/meloxicam, acetaminophen/pentazocine, acetaminophen/clemastine/pseudoephedrine, aspirin/butalbital/caffeine, acetaminophen/caffeine/dihydrocodeine, aspirin/hydrocodone, aspirin/oxycodone, acetaminophen/aspirin/caffeine, aspirin/pravastatin, acetaminophen/dexbrompheniramine/pseudoephedrine, aspirin/meprobamate, aspirin/caffeine/propoxyphene, aspirin/butalbital/caffeine/codeine, aspirin/caffeine/dihydrocodeine, STP707, chlorpheniramine/ibuprofen/pseudoephedrine, licofelone, menatetrenone, polmacoxib, cotsiranib, enflicoxib, icosapent, ECP-1014, aspirin/caffeine/phenacetin, suprofen, lornoxicam, tiaprofenic acid, lumiracoxib, tenoxicam, naproxen/sumatriptan, apricoxib, parecoxib, ibuprofen/phenylephrine, acetaminophen/aspirin/codeine, esomeprazole/naproxen, aspirin/esomeprazole, aspirin/dipyridamole/telmisartan, famotidine/ibuprofen, aspirin/dabigatran etexilate, diclofenac/omeprazole, chlorpheniramine/ibuprofen/phenylephrine, dexamethasone/pomalidomide, sulindac/tamoxifen, sulindac/toremifene, raloxifene/sulindac, ketorolac/phenylephrine, aspirin/bivalirudin, diclofenac/hyaluronic acid, aspirin/clopidogrel, aspirin/omeprazole, aspirin/enoxaparin, aspirin/lisinopril, COX2 inhibitor, diclofenac/misoprostol, acetaminophen/butalbital/caffeine, hydrocodone/ibuprofen, acetaminophen/hydrocodone, acetaminophen/tramadol, acetaminophen/codeine, acetaminophen/oxycodone, acetaminophen/propoxyphene, niflumic acid, nitroaspirin, ketoprofen, diclofenac, etoricoxib, naproxen, meclofenamic acid, pomalidomide, meloxicam, celecoxib, ibuprofen/pseudoephedrine, diphenhydramine/ibuprofen, dipyrone, nimesulide, acetaminophen, mefenamic acid, bortezomib/dexamethasone/pomalidomide, diflunisal, ibuprofen, GW406381X, phenylbutazone, indomethacin, sulfasalazine, SB203580, piroxicam, valdecoxib, aspirin, carprofen, zomepirac, rofecoxib, sorafenib/sulindac/sunitinib, aspirin/caffeine/orphenadrine, acetaminophen/butalbital, balsalazide, aspirin/dipyridamole, acetaminophen/butalbital/caffeine/codeine, naproxen/pseudoephedrine, acetaminophen/diphenhydramine/prednisolone, acetaminophen/diphenhydramine/methylprednisolone, acetaminophen/diphenhydramine, acetaminophen/cetirizine/prednisolone, racemic flurbiprofen, phenacetin, sulindac, nabumetone, etodolac, tolmetin, amlodipine/celecoxib, aspirin/prasugrel, ketorolac, oxaprozin, mesalamine, salsalate, fenoprofen, salicylic acid, aspirin/rivaroxaban, aspirin/clopidogrel/rivaroxaban, aspirin/cangrelor, aspirin/rivaroxaban/ticlopidine, aspirin/ticagrelor, deracoxib, firocoxib, acetaminophen/ibuprofen, acetaminophen/caffeine/chlorpheniramine/hydrocodone/phenylephrine, and bromfenac
RIGI	RNA sensor RIG-I	MK-4621
RIPK1	receptor interacting serine/threonine kinase 1	eclitasertib, GDC-8264, GSK2982772, GSK3145095, and GSK963
RIPK3	receptor interacting serine/threonine kinase 3	N-[6-[3-[(3-bromophenyl)carbamoylamino]-4-fluorophenoxy]-1,3-benzothiazol-2-yl]cyclopropanecarboxamide, GSK843, GSK872, and GSK840
SERPINE1	serpin family E member 1	TM5614, drotrecogin alfa, and ACT001
SIGMAR1	sigma non-opioid intracellular receptor 1	caffeine/dextromethorphan/losartan/midazolam/omeprazole, acetaminophen/pentazocine, dihydrocodeine, dextromethorphan/morphine, dimemorfan, morphine/naltrexone, opipramol, dextromethorphan/quinidine, naloxone/pentazocine, bupropion/naltrexone, naltrexone/oxycodone, bupropion/dextromethorphan, etorphine, SA 4503, fenfluramine, hydromorphone, naltrexone, dextromethorphan, oxycodone, pentazocine, naloxone, SR 31747, brompheniramine/dextromethorphan/pseudoephedrine, chlorpheniramine/dextromethorphan/phenylephrine, carbinoxamine/dextromethorphan/pseudoephedrine, and dextromethorphan/promethazine
STAT3	signal transducer and activator of transcription 3	CAS3/SS3, KT-333, golotimod, OPB-31121, OPB-51602, danvatirsen, TTI-101, STAT3 inhibitor, and NT219
STING1	stimulator of interferon response cGAMP interactor 1	CDK-002, E7766, dazostinag, SNX281, KL340399, ulevostinag, MIW815, GSK3745417, BMS-986301, MK-2118, SB 11285, IMSA101, and BI 1387446
TBK1	TANK binding kinase 1	MRT-68601, 6-aminopyrazolopyrimidine derivative compound II, and BX-795
TGFB1	transforming growth factor beta 1	SHR1701, HB-002T, STP707, cotsiranib, dalantercept, LY2109761, fresolimumab, LY3200882, MSB0011359C, NIS793, AVID200, YL-13027, SRK-181
TLR3	toll like receptor 3	rintatolimod, and poly rI:rC-RNA
TLR7	toll like receptor 7	SHR2150, APR003, BDB018, enpatoran, vesatolimod, UC-1V150, PF-4878691, 5-fluorouracil/imiquimod, resiquimod, hydroxychloroquine, imiquimod, NKTR-262, LHC165, DSP-0509, BDC-1001, TQ-A3334, BNT411, RO7119929
TP53	tumor protein p53	PC14586, eprenetapopt, cenersen, ALT-801, CGM097, kevetrin, azurin 50–77, COTI-2, and BI 907828
TYK2	tyrosine kinase 2	deuruxolitinib, delgocitinib, ropsacitinib, zasocitinib, VTX958, momelotinib metabolite M21, tofacitinib, ruxolitinib, momelotinib, baricitinib, filgotinib, oclacitinib, SAR-20347, brepocitinib, deucravacitinib, baricitinib/methotrexate, and JAK inhibitor I

## Data Availability

Data analyzed in this study are publicly accessible in the GEO database (https://www.ncbi.nlm.nih.gov/geo/, accessed on 16 May 2025) with accession numbers GSE154613, GSE156754, GSE147507, and GSE17400. The Python code for creating the model in the study is available through the following links: Convolutional neural network: https://github.com/Shihori/AI/blob/68bbeaf7944ff8d1c3bbcb40fbf011c853cf4383/CNN_corona2_GitHub.ipynb (accessed on 16 May 2025). Graph neural network: https://github.com/Shihori/AI/blob/018d5cfa90265cf00c6b5cf2a66cd8a2a8009b50/GNN_corona2_GitHub_py.ipynb (accessed on 16 May 2025).

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
