# Peer review of "Artificial Intelligence Approach in Machine Learning-Based Modeling and Networking of the Coronavirus Pathogenesis Pathwayâ€"

_cimb, 2025, doi:10.3390/cimb47060466_

Round 1
Reviewer 1 Report
Comments and Suggestions for Authors
1.The current manuscript exhibits a high degree of similarity to the conference paper in terms of model architecture, data sources, and result presentation, which may not meet the criteria for "substantial extension." As a reviewer, it is difficult to make a clear judgment on this matter.
2. If the conference publisher (such as IEEE or ACM) imposes restrictions on secondary publication, it is necessary to verify whether authorization is required or whether the manuscript meets the criteria for "significant extension."
3.The manuscript claims that the predictive model may aid in the identification of therapeutic strategies, but it does not demonstrate how the model is connected to specific therapeutic targets or drugs.
4.Traditionally, the activation state of a pathway is typically determined based on differential expression analysis or changes in signal intensity. Using "images" to predict pathway activation lacks sufficient theoretical justification.
5.Although IPA is a widely used pathway analysis tool, its data sources and algorithms are proprietary, which may introduce potential biases.
6. No drug screening, drug-pathway matching, or mechanistic inference was performed; drawing therapeutic conclusions solely based on predicted activation states is an overinterpretation.
Author Response
Thank you very much for your constructive comments and for taking time to review our manuscript. Your expertise is greatly appreciated in helping us improve our manuscript.
- The current manuscript exhibits a high degree of similarity to the conference paper in terms of model architecture, data sources, and result presentation, which may not meet the criteria for "substantial extension." As a reviewer, it is difficult to make a clear judgment on this matter.
Response:
Thank you very much for taking the time to review our manuscript very carefully. Upon contacting the Japanese Society for Artificial Intelligence, it was confirmed that the content of the conference paper in Japanese could be prepared as a final paper in English and submitted to other societies or journals. Please refer to 2. “If the Author submits the Paper in its final form to any other academic society, etc., Society shall not object to the Author’s submissions or to that academic society’s use of Paper, etc., on the ground that the Society retains ownership of the Paper” in the Article 4 Grant the Right to Use the Paper of the Copyrights Regulations on Papers Submitted to the Annual Conferences of the Japanese Society for Artificial Intelligence in the links. The links for the English translation and the Japanese original version are as follows:
https://www.ai-gakkai.or.jp/pdf/national-convention/AC_copyright_en.pdf (English translation of the Japanese original version)
https://www.ai-gakkai.or.jp/pdf/national-convention/AC_copyright_jp.pdf (Japanese original version)
Importantly, the conference paper (in 2 pages) was written in the Japanese language, which serves significance in the current manuscript, which was prepared in English and exceeds 20 pages.
- If the conference publisher (such as IEEE or ACM) imposes restrictions on secondary publication, it is necessary to verify whether authorization is required or whether the manuscript meets the criteria for "significant extension."
Response:
Thank you again for this critical comment. Upon contacting the Japanese Society for Artificial Intelligence, it was confirmed that authorization is not required (please refer to the response to your comment 1).
- The manuscript claims that the predictive model may aid in the identification of therapeutic strategies, but it does not demonstrate how the model is connected to specific therapeutic targets or drugs.
Response:
Thank you for your insightful comment. To demonstrate how the model is connected to specific therapeutic targets or drugs, “Using the Grad-CAM model may be helpful for identifying the specific therapeutic targets of drugs.” was added in discussion, lines 349-350 in the revised manuscript. The specific therapeutic target “infected with new coronaviruses” was added in conclusion, line 389.
- Traditionally, the activation state of a pathway is typically determined based on differential expression analysis or changes in signal intensity. Using "images" to predict pathway activation lacks sufficient theoretical justification.
Response:
Thank you so much for your critical comment. The activation state of a pathway is determined based on changes in signal intensity in the current study. The purpose of using “images” colored by changes in signal intensity and predicted activation state is that humans can easily interpret the pathway images with colors. The ultimate goal is to predict the activation state of a pathway based on changes in the expression of nodes in the path. These phrases were added in discussion, lines 350-354.
- Although IPA is a widely used pathway analysis tool, its data sources and algorithms are proprietary, which may introduce potential biases.
Response:
Thank you very much for your insightful comment. Although IPA is a widely used pathway analysis tool, the data sources and algorithms are proprietary, which may introduce potential biases in the prediction models created. Using other pathway analysis tools, such as Reactome or KEGG, would be a future investigation [30,31]. The phrases were added in discussion, lines 359-362.
- No drug screening, drug-pathway matching, or mechanistic inference was performed; drawing therapeutic conclusions solely based on predicted activation states is an overinterpretation.
Response:
Your excellent comments are highly appreciated. Drug-pathway matching was performed and new Table 3 and Figure 4 were added in the revised manuscript.

Reviewer 2 Report
Comments and Suggestions for Authors
Line 70, regarding COVID-19 vaccinations, the authors should cite these studies and specify that in order to ascertain a side effect it is necessary to retrace the WHO guidelines adapted for COVID-19.
1) Sessa, F., Salerno, M., Esposito, M., Di Nunno, N., Zamboni, P. and Pomara, C., 2021. Autopsy findings and causality relationship between death and COVID-19 vaccination: a systematic review. Journal of clinical medicine, 10(24), p.5876.1)
2) Pomara, C., Sessa, F., Ciaccio, M., Dieli, F., Esposito, M., Giammanco, G.M., Garozzo, S.F., Giarratano, A., Prati, D., Rappa, F. and Salerno, M., 2021. COVID-19 vaccine and death: causality algorithm according to the WHO eligibility diagnosis. Diagnostics, 11(6), p.955.
The paper describes the use of an "Artificial Intelligence-based modeling" but the specific details of the Machine Learning algorithm employed (e.g. type of neural network, decision trees, SVM, etc.) and its architecture are insufficiently described. Without this information, it is difficult for other researchers to replicate the study or evaluate its robustness.
Training dataset: It is unclear how the dataset used to train the model was constructed. The “Pathogenesis Pathway” is mentioned but the exact source, size, composition and selection criteria of the data (e.g. gene expression data, protein-protein interactions, clinical data) are vaguely specified.
Materials and methods
The model validation methodology (e.g. cross-validation, split between training/test sets) is not fully explained. Clear performance metrics (e.g. accuracy, precision, recall, F1-score, AUC) are not presented to objectively assess the predictive effectiveness of the model.
Overambitious and unrealistic results:
Discussion
The title and introduction suggest an in-depth analysis of the “Coronavirus Pathogenesis Pathway” through AI. However, the results presented seem to be limited to identifying potential triggers and targets based on existing knowledge or hypotheses, without a clear demonstration of novel predictive or mechanistic findings validated experimentally.
It is unclear how the AI ​​model actually “discovered” or “predicted” novel interactions or pathogenic mechanisms that were not already known or hypothesized. The model output seems more like an organization or restatement of existing data rather than a true AI-driven discovery.
Limited or absent experimental support:
A major challenge for a paper proposing an AI/ML model is the lack of experimental validation of the predicted results. If the model identifies novel pathways or targets, it would be crucial to accompany these predictions with experimental evidence (e.g. in vitro or in vivo experiments) that confirm their biological validity. The complete absence of such validation severely weakens the conclusions.
The paper seems to stop at the computational stage, without linking the results to concrete and verifiable biological implications.
Author Response
Thank you very much for your constructive comments and for taking time to review our manuscript. Your expertise is greatly appreciated in helping us improve our manuscript.
- Line 70, regarding COVID-19 vaccinations, the authors should cite these studies and specify that in order to ascertain a side effect it is necessary to retrace the WHO guidelines adapted for COVID-19.
1) Sessa, F., Salerno, M., Esposito, M., Di Nunno, N., Zamboni, P. and Pomara, C., 2021. Autopsy findings and causality relationship between death and COVID-19 vaccination: a systematic review. Journal of clinical medicine, 10(24), p.5876.1)
2) Pomara, C., Sessa, F., Ciaccio, M., Dieli, F., Esposito, M., Giammanco, G.M., Garozzo, S.F., Giarratano, A., Prati, D., Rappa, F. and Salerno, M., 2021. COVID-19 vaccine and death: causality algorithm according to the WHO eligibility diagnosis. Diagnostics, 11(6), p.955.
Response:
Thank you very much for your insightful suggestion. The suggested references were cited in lines 70-75 in the revised manuscript:
Fatal adverse effects related to COVID-19 vaccines were investigated, which found that autopsy was very useful in defining the main characteristics of the vaccine-induced immune thrombotic thrombocytopenia after ChAdOx1 nCoV-19 vaccination [13]. Causality assessment of adverse events following immunization and COVID-19 vaccination is necessary to retrace the WHO guidelines adapted for COVID-19 [14].
- The paper describes the use of an "Artificial Intelligence-based modeling" but the specific details of the Machine Learning algorithm employed (e.g. type of neural network, decision trees, SVM, etc.) and its architecture are insufficiently described. Without this information, it is difficult for other researchers to replicate the study or evaluate its robustness.
Training dataset: It is unclear how the dataset used to train the model was constructed. The “Pathogenesis Pathway” is mentioned but the exact source, size, composition and selection criteria of the data (e.g. gene expression data, protein-protein interactions, clinical data) are vaguely specified.
Materials and methods
The model validation methodology (e.g. cross-validation, split between training/test sets) is not fully explained. Clear performance metrics (e.g. accuracy, precision, recall, F1-score, AUC) are not presented to objectively assess the predictive effectiveness of the model. Overambitious and unrealistic results:
Response:
Thank you so much for your time for reviewing the manuscript and your comments. The Machine learning algorithm employed is convolutional neural network modeling as described in Materials and Methods 2.5 Python Coding line 166 in the revised manuscript. The detail of the coronavirus pathogenesis pathway is described in Materials and Methods 2.1 Coronavirus Pathogenesis Pathway and the Activation Z-score, and the size and criteria of the data are described in Materials and Methods 2.4 Activity Plot Analysis. A sentence “The details of training data are described in 2.5. Python Coding.” was added in lines 164-165. A sentence “The model validation methodology is based on a split between training and test datasets.” was added in lines 178-179. A sentence “The accuracy of the model tested with 10 datasets was 0.3 before transfer learning, which increased to 0.65 after transfer learning.” was added in lines 294-295. ROC curve of the prediction result of the test data was added in Figure 5b.
- Discussion
The title and introduction suggest an in-depth analysis of the “Coronavirus Pathogenesis Pathway” through AI. However, the results presented seem to be limited to identifying potential triggers and targets based on existing knowledge or hypotheses, without a clear demonstration of novel predictive or mechanistic findings validated experimentally.
It is unclear how the AI ​​model actually “discovered” or “predicted” novel interactions or pathogenic mechanisms that were not already known or hypothesized. The model output seems more like an organization or restatement of existing data rather than a true AI-driven discovery. Limited or absent experimental support:
A major challenge for a paper proposing an AI/ML model is the lack of experimental validation of the predicted results. If the model identifies novel pathways or targets, it would be crucial to accompany these predictions with experimental evidence (e.g. in vitro or in vivo experiments) that confirm their biological validity. The complete absence of such validation severely weakens the conclusions.
The paper seems to stop at the computational stage, without linking the results to concrete and verifiable biological implications.
Response:
Thank you very much for your insightful comments. The sentences “The mechanistic validation of the predicted results is essential for the application of the AI-based machine learning model in clinical use. It would be crucial to have experimental evidence to confirm the biological validity of the novel targets of the coronavirus pathogenesis pathway.” were added in discussion, lines 362-365.

Reviewer 3 Report
Comments and Suggestions for Authors
The authors established a method to predict Coronavirus Pathogenesis Pathways using machine learning by expressing activation and inhibition information on a reference signaling pathway as an image. This approach shows the possibility of obtaining high prediction efficiency by additionally training a model that machine-learns other images using images that represent changes in the signaling pathway. This approach is also valuable because it does not require the construction of complex models and neural networks. However, the reviewer believes that a comparison with a graph neural network should be made to demonstrate the significance of machine learning with images.
Author Response
Thank you very much for your constructive comments and for taking time to review our manuscript. Your expertise is greatly appreciated in helping us improve our manuscript.
Comment:
The authors established a method to predict Coronavirus Pathogenesis Pathways using machine learning by expressing activation and inhibition information on a reference signaling pathway as an image. This approach shows the possibility of obtaining high prediction efficiency by additionally training a model that machine-learns other images using images that represent changes in the signaling pathway. This approach is also valuable because it does not require the construction of complex models and neural networks. However, the reviewer believes that a comparison with a graph neural network should be made to demonstrate the significance of machine learning with images.
Response:
Thank you very much for your constructive comments. A comparison with a graph neural network was made. The code of graph neural network created by Google Gemini is available at the GitHub link: https://github.com/Shihori/AI/blob/018d5cfa90265cf00c6b5cf2a66cd8a2a8009b50/GNN_corona2_GitHub_py.ipynb
The GitHub link for the convolutional neural network and Grad-CAM model with images was replaced with the newest code in the revised manuscript:
https://github.com/Shihori/AI/blob/68bbeaf7944ff8d1c3bbcb40fbf011c853cf4383/CNN_corona2_GitHub.ipynb
The following sentences were added to Discussion in lines 354-358:
The graph neural network model had lower accuracy compared to the convolutional neural network model, highlighting the significance of machine learning with images. Data augmentation by modifying the location of nodes in pathway images may also be a future investigation to enhance the accuracy of the model.

Round 2
Reviewer 1 Report
Comments and Suggestions for Authors
It can be accepted in the current form
Reviewer 2 Report
Comments and Suggestions for Authors
the authors have correctly responded to the reviewers. the paper is ready for publication